# Afrotropical montane birds experience upslope shifts and range contractions along a fragmented elevational gradient in response to global warming

Montague H. C. Neate-Clegg[1]*, Simon N. Stuart[2,3,4], Devolent Mtui[5], Çağan H. Şekercioğlu[1,6], William D. Newmark[7]

1 School of Biological Sciences, University of Utah, Salt Lake City, Utah, United States of America, 2 Synchronicity Earth, London, United Kingdom, 3 A Rocha International, London, United Kingdom, 4 IUCN SSC, David Attenborough Building, Cambridge, United Kingdom, 5 Tanzania Wildlife Research Institute, Arusha, Tanzania, 6 Faculty of Sciences, Koç University, Rumelifeneri, Istanbul, Turkey, 7 Natural History Museum of Utah, University of Utah, Salt Lake City, Utah, United States of America

* monte.neate-clegg@utah.edu

**Data Availability Statement:** All of the historical data have been uploaded to the following Tanzania Wildlife Research Institute publicly accessible

## Abstract

Global warming is predicted to result in upslope shifts in the elevational ranges of bird species in montane habitats. Yet few studies have examined changes over time in the elevational distribution of species along fragmented gradients in response to global warming. Here, we report on a resurvey of an understory bird community in the Usambara Mountains in Tanzania, along a forested elevational gradient that has been fragmented over the last 200 years. In 2019, we resurveyed seven sites, ranging in elevation from 360 m to 2110 m, that were originally surveyed between 1979 and 1981. We calculated differences in mean elevation and lower and upper range limits for 29 species between the two time periods and corrected for possible differences in elevation due to chance. Over four decades, we documented a significant mean upslope shift across species of 93 m. This shift was smaller than the 125 m expected shift due to local climate warming. Of the 29 focal species, 19 shifted upslope, eight downslope, and two remained unchanged. Mean upslope shifts in species were driven largely by contracting lower range limits which moved significantly upslope on average across species by 183 m, while upper range limits shifted non-significantly upslope by 72 m, leading to a mean range contraction of 114 m across species. Community composition of understory bird species also shifted over time, with current communities resembling communities found historically at lower elevations. Past forest fragmentation in combination with the limited gap-crossing ability of many tropical understory bird species are very likely important contributory factors to the observed asymmetrical shifts in lower and upper elevational range limits. Re-establishing forested linkages among the largest and closest forest fragments in the Eastern Arc Mountains are critical to permitting species to shift upslope and to reduce further elevational range contractions over time.

database: bionuwai.tawiri.or.tz. The resurvey data are found in S1 Table.

**Funding:** WDN received a JRS Biodiversity Foundation grant (#60708_TAWIRI; jrsbiodiversity.org). MHCNC received a research grant from the University of Utah Global Change & Sustainability Center (environment.utah.edu/programs-projects/student-funding/). The funders had no role in study design, data collection and analysis, decision to publish, or preparation of the manuscript.

**Competing interests:** The authors have declared that no competing interests exist.

## Introduction

Tropical mountains are globally important for biodiversity [1] and host high levels of species richness and endemism [2, 3]. Yet tropical montane species in general are more threatened [4] and sensitive to anthropogenic change than their temperate counterparts as they tend to be more specialized [5], with low dispersal ability [6], and narrow elevational ranges [7, 8]. Consequently many tropical montane species are threatened by climate change [9]. With increasing global temperatures, many bird species are predicted to shift their elevational distributions upslope [10–12] with species restricted to high elevations being particularly prone to extinction [13] as their elevational ranges contract [14].

In addition to climate change, habitat loss and fragmentation impose further pressures on many tropical montane avian communities [15–17] as burgeoning human populations encroach upon forests. The impacts of habitat loss and fragmentation on tropical bird communities are well-documented [18–21]. However the combined impacts of climate change and habitat loss on tropical birds are less well-understood although potentially devastating [22, 23]. There have been far fewer studies of climate change effects on tropical bird species compared to temperate species [24]. Furthermore, most studies of elevational range shifts of bird communities in the tropics have been conducted along elevational gradients in continuous forest [13, 25, 26]. Consequently, there has been little research assessing how climate-induced elevational shifts of species are impacted in fragmented forests.

Across a species' elevational range, changes over time in lower and upper elevational range limits are a result of range contractions and expansions. Upslope shifts in the lower elevational range of a species represent a population contraction along the trailing edge of a species' range [27], with areas that were formerly suitable becoming unsuitable. On the other hand, upslope shifts in the upper elevational range limit represent a range expansion and necessitate colonization. Although new areas at higher elevations may over time become ecologically suitable for a species, such areas may not be colonized if individuals are unable to reach them [11, 28, 29]. If upslope shifts in the lower elevational range limit exceed upslope shifts in the upper elevational range limit this results in a range contraction.

Forest fragmentation can alter elevational range shifts of species in response to global warming through a restriction of species dispersal. For species with low dispersal capability, upslope shifts in response to climate change will very likely be more constrained in fragmented than in continuous forested landscapes. Many tropical understory bird species have very low dispersal and gap-crossing ability [6, 30, 31] compared to their temperate counterparts. Consequently, colonizing newly-suitable habitats at higher elevations separated by forest gaps may be particularly problematic for many understory bird species. As a result upper elevational range limits for many tropical understory bird species may shift minimally in response to global warming while lower range boundaries may contract leading to a compression in elevational ranges [13, 14]. Alternatively, forest fragmentation may reduce competitively-driven upslope shifts of species due to the inability of lowland species to disperse to higher elevation fragments [32].

Most research on elevational range shifts of bird species over time in the tropics has been conducted in the Neotropics [13, 16, 25, 33] and Southeast Asia [15, 26, 34]. There has been relatively little research in the Afrotropics (but see: [35–38]). This is salient because most montane forests in Africa are isolated from one another and continuous elevational gradients have often received incomplete protection [39, 40]. For example, the 13 Eastern Arc Mountains of Tanzania and Kenya are separated on average by tens to hundreds of kilometers [31]. Such distances preclude dispersal of forest-dependent species with low dispersal ability [17, 21, 41]. Furthermore, within the Eastern Arc Mountains, forests are extensively fragmented [31]

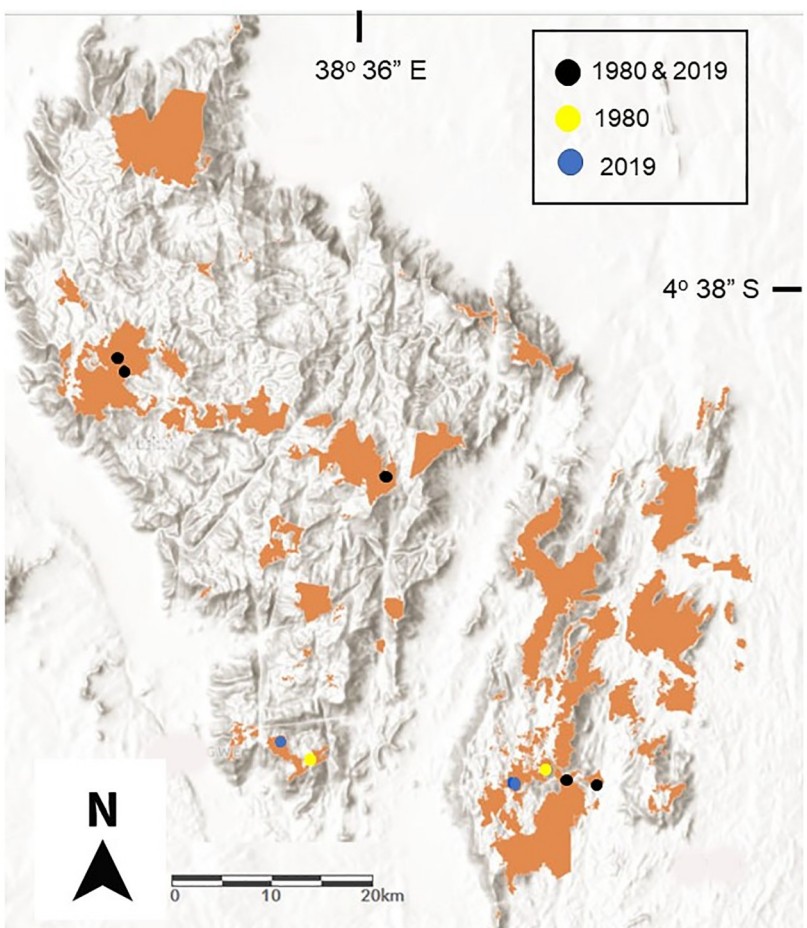

**Fig 1. Natural forest cover and location of 1980 and 2019 survey sites in the East and West Usambara Mountains in northeast Tanzania.**

(Fig 1). While there is some evidence that climate change may be altering the elevational distribution of Afrotropical birds in montane habitats [37], there has been little assessment of elevational shifts of species over defined time periods.

Here, we repeat a historical elevational transect of understory bird species conducted between 1979 and 1981 along a fragmented gradient in the East and West Usambara Mountains in northeast Tanzania (Fig 1). We assess shifts over time in the elevational distributions of species by calculating range shifts of species' mean elevations as well as shifts in species' lower and upper range limits. We then compare observed shifts to expected shifts due to climate warming. Finally, we examine changes over time in community composition along an elevational gradient. In the absence of habitat fragmentation, we would expect species' elevational ranges to shift upslope over time. However, as a result of habitat fragmentation in the East and West Usambara Mountains, and based on observed elevational range shifts of species in these mountains, we predict that lower elevational range limits should shift upslope faster than upper elevational range limits resulting in a range contraction across species in comparison to tropical montane species occurring in continuously forested landscapes (Fig 2).

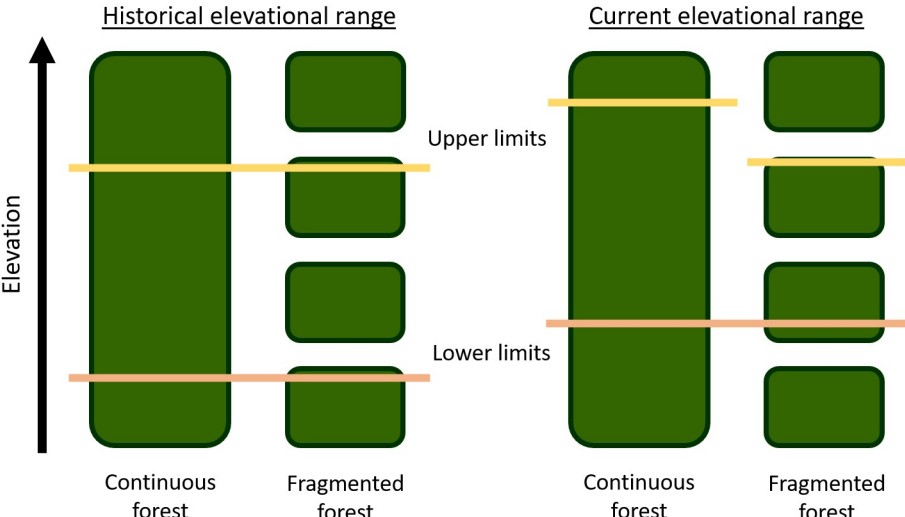

**Fig 2. Predicted elevational range shifts of species over time in response to global warming in continuous versus fragmented forest.** Yellow and pink lines illustrate the upper and lower elevational range limits of a species, respectively. In continuous forest upslope shifts in lower and upper elevational range limits of species should be approximately equivalent. In fragmented forest containing species with limited dispersal ability, climate warming is predicted to result in upslope shifts of lower elevational range limits exceeding upslope shifts in upper elevational range limits resulting in a compression of the elevational range of species.

## Materials and methods

### Study sites

This study was conducted in the East and West Usambara Mountains in northeastern Tanzania which are part of the Eastern Arc Mountains, a global biodiversity hotspot [1]. Over the last two centuries the Eastern Arc Mountains have experienced a 77% loss in original forest cover [31]. The remaining forests in the Eastern Arc Mountains, including forests in the East and West Usambara Mountains, are highly fragmented. The East Usambara consist of 31 fragments >10 ha in size and the West Usambara consist of 38 fragments >10 ha [31] (Fig 1). Forest fragments in the Usambara Mountains are surrounded by a matrix of varying land use consisting predominantly of small-scale agriculture in combination with large-scale agriculture (predominantly tea, and sisal at lower elevations), and timber and fuel wood plantations (teak, *Eucalyptus*, pine). Importantly, based on a comparison of historical Google Earth aerial imagery having a spatial resolution of 30 m (Landsat/Copernicus) with current imagery having a spatial resolution of 0.5–1.5 m, the area of the five surveyed forest blocks (see below) has not changed between 1979 and 2019 (S1 Fig). Matrix habitat composition over the last four decades within 500 m of the forest edge has changed minimally (Table 1; S1 Fig). The largest change by area in matrix habitat composition was a conversion between 2005–2017 of small-scale agriculture to a pine plantation adjacent to the highest elevation forest block.

The difference in mean daily temperature between the cool and hot seasons (see below) recorded over a 22-month period at the 1050 and 1310 m study sites is 3.86°C [39]. Rainfall in the East and West Usambara Mountains is bimodal with the long rains extending from March–June and the short rains running from October–December [39].

**Table 1. Description of seven study sites along an elevational gradient in the East and West Usambara Mountains of Tanzania.**

| Location | Elevation of study site (m) | Forest block area (ha) | Distance to next higher surveyed forest block (km) | Change by area (1979–2019) in adjacent matrix habitat composition (%) | Latitude | Longitude | Length of mist net line (m) |
|---|---|---|---|---|---|---|---|
| East Usambara | 360 | 7018† | .03 | <1 | -5.1118 | 38.6784 | 100 |
| East Usambara | 580 | 7018† | .03 | <1 | -5.1008 | 38.6570 | 190 |
| East Usambara | 1020 | 640 | 14.2 | <1 | -5.1058 | 38.5979 | 380 |
| West Usambara | 1310 | 908 | 24.7 | <1 | -5.0664 | 38.4110 | 380 |
| West Usambara | 1530 | 3690 | 16.6 | <1 | -4.8219 | 38.5051 | 380 |
| West Usambara | 1820 | 4898† | - | 9 | -4.7328 | 38.2681 | 380 |
| West Usambara | 2110 | 4898† | - | 9 | -4.7121 | 38.2442 | 380 |

† Located in the same forest block.

## Bird surveys

Between July and September 2019 we repeated a historical survey that was originally conducted between April and September (the cool season) in 1979, 1980, and 1981 at sites that ranged in elevation from 360 m to 2110 m (Table 1), spaced at approximately 300 m elevational intervals [42]. Survey sites were located in five large forest blocks that ranged in size from 640 ha to 7018 ha, and distance between adjacent surveyed forest blocks ranged from 0.03–24.7 km (Table 1). The elevation of survey sites was derived from a Digital Elevation Model (DEM) for Tanzania.

In the 2019 resurvey, we replicated historical sampling methods. At each location, we erected a single line of mist nets (12m x 2 m, 36 mm mesh) of equivalent length (100 m, 190 m, and 380 m; Table 1) as conducted in the historical survey. All mist net lines ran perpendicular to the forest edge and began 100 m from the forest edge, with the exception of the mist net line at the 360 m elevation site that began at the edge. Mist net lines were operated from dawn to dusk (12 hours) over five continuous days and all captured birds were banded and standard morphometric measurements recorded. Although the historical survey was conducted over a ten-day period at each location, because of the high drop-off in capture rate over time, the resurvey effort was limited to five days per site. Consequently, we restricted the analysis of the historical survey data to the first five days of that survey. Permission for this study was granted by the Tanzania Wildlife Research Institute and the Tanzania Commission for Science and Technology (permit #2019–448 NA-2019-302).

## Analysis

Following Forero-Medina et al [25], we restricted the analysis to species caught at least twice during both the historical and current resurvey, because a minimum of two captures are required to calculate elevational range. For each time period and species, we calculated mean elevation, and the 2.5% and 97.5% elevational range quantiles to define the "lower" and "upper" range limits, respectively. Percent quantiles are preferable to the minimum and maximum values of elevation in assessing lower and upper range limits because they are less influenced by individual outliers and incorporate all data. In calculating a species' elevational shift

over time, we subtracted historic elevation values from current values. In calculating upper range limit shifts across species, we excluded all species that were recorded at the highest elevation site (2110 m) during both surveys because for these species it is not possible to detect upslope shifts in upper range limits [13]. Similarly, in calculating lower range limit shifts we excluded all species that were recorded at the lowest elevation site (360 m) during both surveys. We did, however, incorporate species' maximum and minimum recorded elevational range records in each of the two surveys in assessing number and proportion of elevational range shifts that occurred within continuous versus fragmented forest because the 2.5% and 97.5% elevational range quantiles are calculated rather than observed values. For those species whose maximum or minimum recorded elevation were higher in 2019 than 1980, we calculated the proportion of species for which this change occurred within contiguous versus fragmented forest.

We then assessed across species whether shifts differed significantly from 0 using a weighted two-tailed t-test ("wtd.t.test" from package *weights*) weighted by log number of captures. We used a weighted t-test rather than a conventional t-test because species' sample sizes varied widely ranging from 2 to 84 captures. Estimates of elevational range shifts for species with large sample sizes are more reliable [43–45] and hence are given greater weight under a weighted t-test, but log-transforming sample sizes ensured that the t-statistics were not excessively weighted towards a few species with large sample sizes [46].

However, a detection of a difference in elevation between two time periods may be spurious, particularly if numbers of captures of individuals differ between time periods. We therefore conducted Monte Carlo simulations to generate differences in elevational range expected by chance alone [25]. In this simulation, we first pooled captures of all species across both time periods for each site, and then randomly assorted captures to each time period, keeping total captures during each period constant. For example, if 100 individual birds were captured in the historical survey at the 1020 m site, and 60 individuals were captured there in the resurvey, we combined captures from the two periods and then randomly selected without replacement 100 and 60 individuals from this combined pool to create a sample. From this randomized dataset, we calculated the difference in mean elevation and upper and lower range limits between the two periods for each species as above. We repeated this process 1000 times to construct a distribution of expected differences in elevation by species and then calculated the mean and 95% confidence intervals for these simulated distributions. If observed shifts fell outside the 95% confidence intervals of the expected differences, then a species was deemed to have shifted. We then calculated corrected shifts by subtracting expected shifts from observed shifts. Finally, we conducted a weighted two-tailed t-test on corrected shifts to assess whether they differed significantly from 0.

To assess how closely bird species were tracking temperature change over time, we first calculated changes in annual mean temperature (1979–2019) derived from monthly mean temperature data available at half-degree grid cell resolution from the Climatic Research Unit, University of East Anglia [47] for the four 0.5˚ x 0.5˚ grid cells that overlap the seven study sites. We then took the average value for these four grid cells for each year and used a linear model to estimate annual temperature change. To calculate the expected distance of upslope shift required to match the increase in local temperature, we combined the annual change in temperature with the known lapse rate of 0.68˚C per 100 m for the Usambara Mountains [39]. We assumed that the expected distance of upslope shift was constant across a species' elevational range.

To assess the changes in community composition over time, we analyzed all species that were recorded during the historical and current surveys, including species that were captured once. For each site and time period, we recorded the number of individuals captured and

conducted a Principal Coordinate Analysis on a Bray-Curtis dissimilarity matrix [48]. This analysis collapses the community composition into the most important axes of variation and, when plotted, points that are closer in Euclidean space are more similar in community composition. The distribution of points (sites) between the two time periods along the first major axis were compared with a paired t-test.

All analyses were conducted in R version 3.6.1 [49].

## Results

### Elevational range shifts

A total of 1207 birds of 54 species were captured across the two surveys comprising 808 birds of 51 species in the historic survey and 399 birds of 44 species in the resurvey, under an equivalent sampling effort. The >50% decline in the total number of birds captured and >13% decline in the number of species captured between the historic and the current resurvey illustrate the community-wide faunal collapse or relaxation that has been occurring across even the largest forest blocks in the Usambara Mountains over the last 40 years [17, 50, 51]. Of 54 species recorded along the elevational transect across both surveys, 29 species were captured at least twice during each survey (S1 Table). The uncorrected (see below) shift across these 29 species (S1 Table) averaged 95.3 ± 88.2 m (95% CI) upslope which is marginally significant ($t_{28}$ = 2.02, $p$ = 0.053). The uncorrected lower elevational range limits across species shifted significantly ($t_{23}$ = 4.04, $p < 0.001$) upslope on average by 258.4 ± 127.5 m. Conversely, the uncorrected upper elevational range limits of species shifted upslope by 5.4 ± 123.6 m which is not significant ($t_{15}$ = -0.12, $p$ = 0.91). Of the six species whose maximum elevation was higher in 2019 than 1980, five were recorded at a higher elevation survey site within the same forest block where they were previously recorded. We recorded only one species, Little Greenbul (*Eurillas virens*), shifting its maximum elevation upslope over time between forest blocks. By contrast, of the 14 species whose minimum elevation was higher in 2019 than 1980, ten of these 14 upslope shifts were recorded across one or more forest blocks.

Detected differences in the elevational range of species could be due to chance, particularly because the numbers of captures differed between the two time periods. Under a null hypothesis that assumes no change in elevational range generated through Monte Carlo simulations, the expected difference in mean elevation across species was +2.8 ± 9.7 m while the expected difference in lower and upper range limits were +75.5 ± 31.0 m and -66.4 ± 35.9 m, respectively. After correcting at the species level for expected differences in elevation due to chance, species significantly ($t_{28}$ = 2.07, $p$ = 0.048) shifted their mean elevational range upslope on average by 92.5 ± 84.9 m (S1 Table), with 19 of 29 species shifting upslope, eight species shifting downslope, and two species remaining unchanged (Fig 3). The lower elevational range limit across species also significantly ($t_{23}$ = 3.46, $p$ = 0.002) shifted upslope on average by 182.8 ± 104.9 m; while the upper elevational range limit shifted upslope by 71.8 ± 117.8 m (Fig 3) which is not significant ($t_{15}$ = 1.21, $p$ = 0.25). If lower elevational range limits shift faster than upper elevational range limits this results in an elevational range contraction. We found that for those species for which it was possible to estimate shifts in both their upper and lower elevational range limit (n = 11), the elevational ranges for these species contracted (lower limit shift—upper limit shift) on average by 114.4 ± 168.5 m.

Finally, we calculated how closely understory bird species were tracking changes in local temperature over time. Between 1979 and 2019 mean annual temperature across the seven study locations increased by 0.022˚C per year which equates to an increase of 0.85˚C over the 39-yr time interval. Combining this increase in temperature with the observed lapse rate in the Usambara Mountains of 0.68˚C per 100 m the expected mean shift across species was 124.7 m.

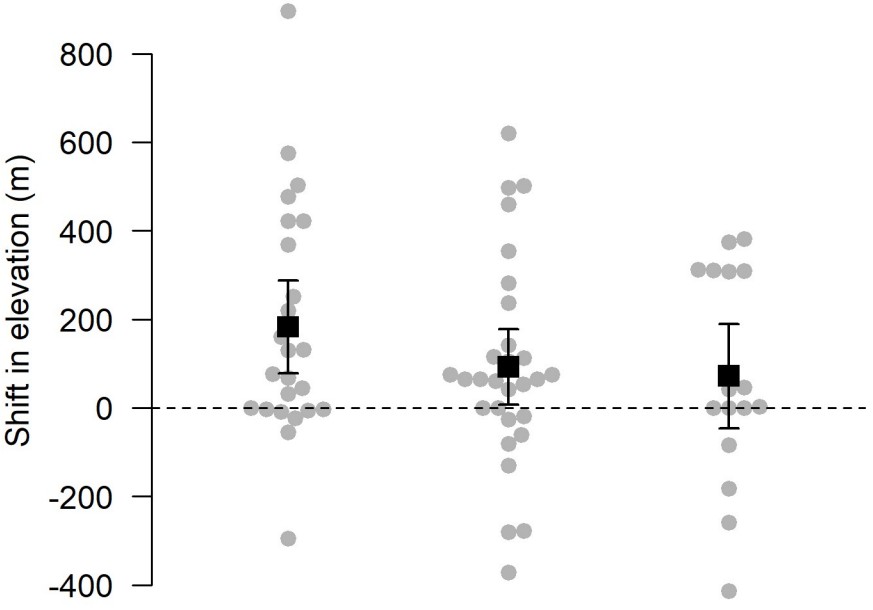

**Fig 3. Shifts in mean elevation and lower and upper elevational range limits of 29 species of understory bird species between 1980 and 2019 in the Usambara Mountains, Tanzania.** Elevational shifts were corrected for differences in elevation expected by chance. Each point represents a species. Black squares and bars represent the mean and 95% confidence intervals across species.

Thus, species shifted on average upslope by 74% (92.5 m/124.7 m), which is lower than expected; their lower elevational limits by 147%, (182.8 m/124.7 m) which is higher than expected; and their upper elevational limit by 58%, (71.8 m/124.7 m) which is lower than expected.

## Community structure

Based on a Principal Coordinate Analysis on a Bray-Curtis dissimilarity matrix, the first two axes explained 59% of the variation in community dissimilarity. The first axis of community composition (Fig 4) accounted largely for the elevational distribution of species, with the community composition of lower elevation sites having lower values along Axis 1. The elevation of sites was positively correlated with Axis 1 values (Pearson's $r = 0.94$, $t = 9.80$, $p < 0.001$). Comparing historic to current community composition using a paired t-test of Axis 1 values indicated that the current community composition had more negative values across all study sites than the historic community composition ($t_6 = -3.67$, $p = 0.010$). Thus, community composition has shifted in a negative direction (to the left) along Axis 1 over time (Fig 4), indicating that the current community composition at higher elevations is resembling the historic community composition at lower elevations (Fig 4).

## Discussion

Here we provide evidence that along a fragmented elevational gradient in Tanzania, understory forest bird species have shifted their mean elevational range upslope over the last four decades in response to climate warming (Fig 3), with concomitant changes in community

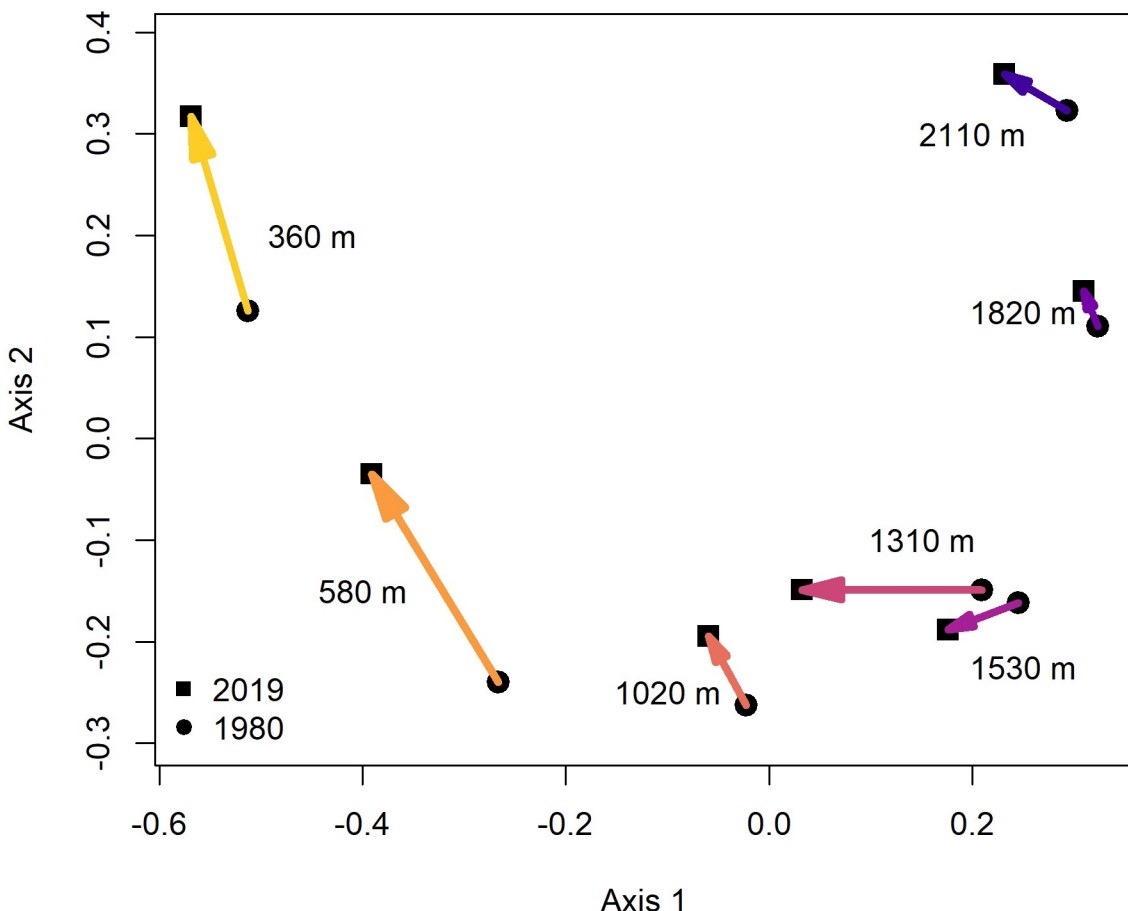

**Fig 4. Shifts in the community composition of understory birds across an elevation gradient between 1980 and 2019 in the Usambara Mountains, Tanzania.** Species composition is based on a Principal Coordinate Analysis of a Bray-Curtis dissimilarity matrix. Closer points are more similar in community composition. Arrows point from the historic to the current community composition at each site.

composition (Fig 4). However, shift rates were faster at species' lower elevational limits than at their upper elevational range limits, resulting in a range contraction across species. We found across 29 species in the East and West Usambara Mountains that the mean elevational range shifted upslope by 93 m over a 39-year period which is equivalent to 2.4 m/yr. Such upslope shifts are consistent with recent findings from the Neotropics [13, 16, 25, 52], the Albertine Rift [38] and Southeast Asia [26, 34]. Although it has been predicted that montane tropical birds, in general, will shift their distributions upslope over time in response to global warming [10], we observed that the understory bird species in the Usambara Mountains shifted their mean elevational ranges upslope by 74%, on average, of the expected distance if bird species were to perfectly track the increases in local temperatures [25].

Mean upslope shifts occurred across species having very different species traits including both common (Shelley's Greenbul, *Aruzelocichla masukuensis*) and uncommon species (Orange Ground-thrush, *Geokichla gurneyi*), edge-tolerant (Little Greenbul, *Eurillas virens*) and edge-averse species (Spot-throat, *Modulatrix stictigula*), terrestrial (Grey-breasted Illadopsis, *Illadopsis distans*) and midstory species (Bar-throated Apalis, *Apalis thoracica*), and species from multiple feeding guilds including insectivores (White-tailed Crested-flycatcher, *Elminia albonotata*), omnivores (Olive-headed Bulbul, *Arizelocichla striifacies*), and granivores (Oriole

Finch, *Linurgus olivaceus*). Thus, there would appear to be few species traits that are associated with upslope shifts across species, although rigorously assessing species traits is problematic due to sample size (n = 29). These results are consistent with a lack of pattern in species traits associated with shifts observed in the Neotropics and Southeast Asia [13, 25, 26], although in the Albertine Rift of Rwanda shift rates in tropical birds have been associated with avian body size [38]. The association between shift rates and species traits clearly warrants further investigation across the tropics.

In this study, of the 27 species that shifted their mean elevational range, eight species (28%) shifted their mean elevational range downslope. This proportion is similar to that observed in the Neotropics and Southeast Asia [13, 25, 26] where one-fifth to one-third of species have been observed to shift their mean elevational ranges downslope. The general consistency of this pattern across the tropics suggests that factors other than increasing temperature *per se* are important drivers of elevational range shifts for many species [53, 54]. Determinants of the elevational range limits of birds are complex [55] and individual species can respond very differently to climate change [56]. Temperature- and precipitation-induced changes in habitat structure [57, 58], food availability [59], natural enemies [60], and competition [32, 61, 62] have all been shown to influence range limits in birds [55]. A better mechanistic understanding of the interplay between these factors [58] is obviously required to more accurately predict the responses of tropical bird species to climate change and comparisons of species responses across the tropics may be useful and should provide additional insights.

In this study, the average rate at which the lower elevational range limit shifted upslope across species (182.8 m, 4.7 m/yr) was 2.5 times faster than that of the upper elevational range limit (71.8 m, 1.8 m/yr). As a result, the elevational ranges of bird species in the Usambara Mountains contracted on average by 114.4 m over the 39-year period. Along continuously forested gradients in New Guinea and Peru, lower elevational range limits of species shifted at a lower rate (New Guinea: 2.0–2.8 m/yr, Peru: 3.7 m/yr), whereas upper elevational range limits of species shifted at a higher rate (New Guinea: 2.4–3.5 m/yr, Peru: 2.1 m/yr) than in the Usambara Mountains, consistent with the prediction of elevational range contractions of bird species in fragmented forest (Fig 2). Some caution is warranted in comparing precise shift rates because of the slightly different methods that were used to calculate lower and upper range limits. Yet most notably in these latter studies, which were conducted in continuous forest, elevational ranges did not contract on average across species unlike in the Usambara Mountains. This pattern is also consistent with predictions of elevational range shifts of species in continuous versus fragmented forest (Fig 2).

Several factors may have contributed to the differential shift rates in lower and upper elevational range limits and resulting range contractions of bird species in the Usambara Mountains. In this study we observed that five of the six (83%) upslope shifts in species' maximum elevations occurred within a forest block (i.e. continuous forest) between 1820 m and 2110 m. In contrast, over the 39-year study period, we documented only one species, Little Greenbul (*Eurillas virens*), shifting upslope over time between forest blocks. The Little Greenbul is an edge-tolerant species [20] that has previously been documented in the Usambara Mountains to readily cross forest gaps [21]. On the other hand, of the 14 species whose minimum elevations shifted upslope, 71% of these observed shifts occurred across one or more forest blocks, indicating a population contraction along the trailing edge of these species' elevational ranges. This difference suggests different processes are impacting species' leading and trailing elevational range boundaries.

In continuous forest, upslope shifts in species' upper and lower range boundaries are unconstrained by forest gaps [14]. In contrast, in fragmented forest, forest gaps can represent a serious barrier to species dispersal. Over the last 200–300 years, forests in the East and West

Usambara Mountains have been highly fragmented by human activities [31, 39] (Fig 1). The mean distance (± SD) between the forest blocks included in this study is 13.9 km ± 10.3 km (Table1; Fig 1). Furthermore, long-term monitoring of understory bird species in the East and West Usambara Mountains has indicated very limited dispersal ability for most species [21, 63]. At the 1020 and 1310 m study sites, understory bird species have been surveyed annually over 34 years across a network of 14 forest fragments [21, 51]. Over this time period and based on >31,500 captures and $4.43 \times 10^6$ m-mistnet-hrs, 21% of all understory bird species have never been recorded within a study fragment other than where they were originally banded [17]. The reluctance and/or inability of these species to cross forest gaps in the Usambara Mountains has been further confirmed through detailed radio-tracking and foraging observations in matrix habitats [21, 64]. Radio-tracking of four understory bird species with negligible gap-crossing ability over an eight-year period recorded no bird ever crossing a non-forested gap >15 m [21, 64]. Limited gap-crossing capacity and dispersal ability have also been observed among many other tropical bird species in other regions [6, 17, 30, 41, 65].

For those bird species in the Usambara Mountains which have been documented to cross forest gaps, mean distance of movement between forest fragments is <150 m assuming a bird 'leap-frogged' across the landscape (which is how radio-tagged birds in translocation experiments in the tropics have been observed moving through matrix habitats) or 360 m assuming a bird flew in a straight line between forest fragments [17]. While selected understory bird species with good dispersal capability [21] may be able to disperse upslope over time by leap-frogging between small forest fragments, the distance between the remaining small forest fragments is still large relative to the dispersal ability of many species. The median distance between forest fragments >10 ha in size in the East and West Usambara Mountains is 581 m and 1387 m, respectively [31]. And even if individual species are capable of dispersing upslope between forest blocks, establishing a population may be inhibited by an absence of attracting conspecifics [66, 67] and/or by Allee effects [68, 69]. Consequently, there are clearly multiple ecological constraints preventing species from shifting their upper elevational ranges upslope in fragmented forest in the East and West Usambara Mountains.

In contrast to upslope shifts in species' upper elevational range limits, upslope shifts in the lower elevational range limits of species in fragmented landscapes are unconstrained by colonization (Fig 2) as they represent the loss of individuals from trailing edge habitats. Nonetheless, it is possible that forest gaps may act as an impediment to lowland competitors and/or natural enemies that may constrain the lower elevational limits of bird populations [55]. For example, if the lower elevational limit of a species is a result of interspecific competition [32, 70], and a competitor is unable to disperse upslope because of a forest gap, this could also alter species shift rates. However, unlike in regions such as the Andes where parapatric species replacements are common [62], in the Usambara Mountains there are very few parapatric species pairs. Thus, there is less evidence to suggest that reduced interspecific competition may be influencing lower limit shift rates. On the other hand, habitat fragmentation will almost certainly exacerbate impacts of global warming on shift rates of understory bird species along their lower elevational range limits via lag effects between past habitat loss and species responses [50] as well as edge effects on habitat selection and nest survival [20, 71]. Fragmentation will therefore almost certainly amplify the impacts of global warming on population declines of tropical montane bird species along their trailing edge [15, 16].

Over the next four decades, annual mean temperatures across Tanzania are projected to increase by 1.0 to 2.7°C [72]. Understory bird species in the Usambara Mountains would consequently be projected to shift their mean elevational range upslope by 114–310 m and their lower elevational range limit upslope by 215–581 m in order to track local temperatures while upper elevational range limits would very likely shift minimally. Given the fragmented nature

of montane forests in the Eastern Arc Mountains and the limited ability of many understory bird species to cross forest gaps, a concerted effort to reconnect the largest and closest forest fragments with reforested linkages will be essential in the Eastern Arc Mountains [17] to reduce future elevational range contractions due to climate change and to permit species to shift their elevational ranges upslope in response to climate change.

The Eastern Arc Mountains, including the East and West Usambara Mountains, are globally important for threatened and endemic species, including birds [73]. In the Usambara Mountains, nine bird species are currently listed as globally threatened: Long-billed Forest Warbler (*Artisornis moreaui*–Critically Endangered (CR)); Sokoke Scops Owl (*Otus ireneae*–Endangered (EN)); Usambara Akalat (*Sheppardia montana*–EN); Usambara Hyliota (*Hyliota usambara*–EN); Amani Sunbird (*Hedydipna pallidigaster*–EN); Usambara Weaver (*Ploceus nicolli*–EN); Banded Sunbird (*Anthreptes rubritorques*–Vulnerable (VU)); Swynnerton's Robin (*Swynnertonia swynnertoni*–VU); and Dapple-throat (*Arcanator orostruthus*–VU). Of these, only one—the Usambara Akalat—was caught in sufficient numbers to be incorporated in the analysis. This species is endemic to the high elevations in the West Usambara Mountains. Forty years ago, it was abundant in the forests >1800 m in elevation, but our results suggest a decline on the order of 60%. As a mountain-top species, it cannot move any further upslope, and its conservation status is therefore of serious concern.

The remaining eight globally threatened species include four canopy species (Usambara Hyliota, Amani Sunbird, Banded Sunbird and Usambara Weaver) which were unlikely to be caught in mist-nets. However, five species (Long-billed Forest Warbler, Sokoke Scops Owl, Usambara Hyliota, Amani Sunbird, Swynnerton's Robin and Dapple-throat) were already very rare in the Usambara Mountains 40 years ago based on both mist-net and line transect surveys [42] and are currently so based on mist-net surveys and opportunistic sightings (WDN; MHCNC personal observations). Consequently, climate change has very likely also impacted the elevational range of these species over the last four decades. There should be particular concern for the Long-billed Forest Warbler and the Usambara Hyliota, both of which are endemic to the East Usambara Mountains and have very narrow elevational ranges. The IUCN Red List status for all of these globally threatened species should be reassessed. We also recommend that conservation plans be developed for all of these species as soon as possible, and these should include a focus on emergency conservation measures, such as reconnecting forest fragments, and possibly even assisted colonization for species that face local or even global extinction without such measures [74].

Our results indicate that over the last four decades lower elevational range limits of understory bird species in the East and West Usambara Mountains have shifted upslope, upper elevational range limits have largely remained stationary, and consequently average elevational ranges of species have contracted. If this is the case for relatively mobile species such as birds, we can only assume that the situation is almost certainly worse for even more sedentary groups of animals and plants. Given the global importance of the East and West Usambara and other Eastern Arc Mountains for biodiversity conservation, a conservation strategy for species that addresses climate change is urgently needed.

## Supporting information

**S1 Table. Historic and current elevational range limits, and observed and corrected elevational range shifts for 29 focal understory bird species in the Usambara Mountains, Tanzania.** Number of captures are across both historic and current surveys.
(DOCX)

**S1 Fig. Time-series Google Earth aerial imagery (1984–2016) of study forest blocks and adjacent matrix habitats in the East and West Usambara Mountains, Tanzania.** The forest block displayed in row 1 contains survey sites at 360 m and 580 m; row 2 contains the survey site at 1020 m; row 3 contains the survey site at 1310 m; row 4 contains the survey site at 1530 m; and row 5 contains survey sites at 1820 m and 2110 m. The perimeter of each forest block is outlined in white, and matrix habitats within 500 m of the forest edge are outlined in red. (PDF)

**S2 Fig.**
(JPG)

# Acknowledgments

We thank the Tanzania Wildlife Research Institute and Tanzania Commission for Science and Technology for permission to conduct this study, and V. Mkongewa, D. Barua, H. Pombekali, B. Antoni, J. Khaba, and J. Ayubu for their assistance in the field.

# Author Contributions

**Conceptualization:** Montague H. C. Neate-Clegg, William D. Newmark.

**Data curation:** Montague H. C. Neate-Clegg, Simon N. Stuart.

**Formal analysis:** Montague H. C. Neate-Clegg.

**Funding acquisition:** Montague H. C. Neate-Clegg, Simon N. Stuart, Devolent Mtui, William D. Newmark.

**Investigation:** Montague H. C. Neate-Clegg, Simon N. Stuart, William D. Newmark.

**Methodology:** Montague H. C. Neate-Clegg, Simon N. Stuart, William D. Newmark.

**Project administration:** Simon N. Stuart, Devolent Mtui, William D. Newmark.

**Resources:** Devolent Mtui, William D. Newmark.

**Supervision:** Çağan H. Şekercioğlu, William D. Newmark.

**Visualization:** Montague H. C. Neate-Clegg.

**Writing – original draft:** Montague H. C. Neate-Clegg.

**Writing – review & editing:** Montague H. C. Neate-Clegg, Simon N. Stuart, Devolent Mtui, Çağan H. Şekercioğlu, William D. Newmark.

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
