## [Decision Letter · Decision Letter 0]

19 Nov 2020

PONE-D-20-31582

Afrotropical montane birds experience upslope shifts and range contractions along a fragmented elevational gradient in response to global warming

PLOS ONE

Dear Dr. Neate-Clegg,

Thank you for submitting your manuscript to PLOS ONE. After careful consideration, we feel that it has merit but does not fully meet PLOS ONE’s publication criteria as it currently stands. Therefore, we invite you to submit a revised version of the manuscript that addresses the points raised during the review process.

We look forward to receiving your revised manuscript.

Kind regards,

Bhoj Kumar Acharya, PhD

Academic Editor

PLOS ONE

Journal Requirements:

**Additional Editor Comments:**

In addition to the comments of the three reviewers, I have few following comments:

1. The MS do not have single table and has only two figures. The data can be more appropriately visualized graphically or through tables. Authors can atleast provide map of the study area showing current and historical sampling locations (hope they are not exactly the same). I understand that sampling elevations are same but I presume locations must be different. Hence, make this point clear in the methods section and include map as suggested.

2. Since there will be definite LULC change of the landscape between historical period (1979-81) and current (2019), I suggest the authors to prepare LULC map of both time scale and report any change that have undergone during this 40 years period. This will be helpful to understand whether some of the range shifts in birds is due to landuse change rather than climate change per se. Otherwise, how did authors control over land use changes (leading to habitat change) over these years?

Reviewers' comments:

Reviewer's Responses to Questions

**Comments to the Author**

1. Is the manuscript technically sound, and do the data support the conclusions?

Reviewer #1: Yes

Reviewer #2: Partly

Reviewer #3: Partly

2. Has the statistical analysis been performed appropriately and rigorously? 

Reviewer #1: Yes

Reviewer #2: Yes

Reviewer #3: Yes

3. Have the authors made all data underlying the findings in their manuscript fully available?

Reviewer #1: Yes

Reviewer #2: Yes

Reviewer #3: Yes

4. Is the manuscript presented in an intelligible fashion and written in standard English?

Reviewer #1: Yes

Reviewer #2: Yes

Reviewer #3: Yes

5. Review Comments to the Author

Reviewer #1: This is a great paper. It clearly describes why the study is important, what the authors found, and what these results mean. I have only one minor comment.

1) Please clarify how elevational shifts were calculated for upper and lower range limits. In particular, it does matter whether all species are included in these calculations or whether species historically found at the lower limit were excluded for lower limit shifts and at upper limit excluded for upper limit shift. This is mentioned in the Discussion but should be incorporated into the more general results as well. Particularly for the upper limits, it makes sense to NOT include species that already live at the highest elevation when calculating upper elevation shifts.

Also please fix the references in text to consistently match the correct style

Reviewer #2: My only concern about this manuscript is with the interpretation that these species are moving upslope rather than just disappearing from lower elevations. The study landscape is a very complex one and the West Usambaras (all of the high-elevation sites) and the east Usambaras (all of the low-elevation sites) are separated by a large lowland savannah/agricultural gap and seem very unlikely to have much bird movement between them. Rather, they seem to have independently evolved communities in which all of the movements are likely to have been within them. I therefore wonder if the threat is not much greater than indicated with the east Usambara species simply being pushed upslope with nowhere to go. All of the other comparable paper have dealt with continuous transects in which dispersal is not limited. If you have evidence to the contrary, then I would need to see it to have enough confidence that these bids can in fact cross these gaps. If not, then the analysis needs to be done separately for the two mountain ranges.

Reviewer #3: Summary:

This study performs a resurvey of bird communities (following on initial studies 40 years ago) along an elevational gradient made up of forest fragments in the East and West Usambara Mountains of Tanzania. The authors document shifts in mean, lower and upper limits for nearly 30 species, using a range of methods including Monte Carlo simulations to test for significant shifts and ordination analysis of community similarity metrics. The majority of species showed upslope shifts, though a handful shifted downslope in this landscape, and lower elevation limits shifted more than upper limits, resulting in range contractions in the community. The authors point to influences such as the limited ability of species to cross gaps, which could impact the observed range shifts in this Afromontane region.

I think this is a solid study that could make a nice contribution to PLoSONE, though with substantive revisions. The dataset is unique and important, the manuscript is well written and the text is balanced. Yet I feel that the manuscript is falling slightly short of the mark, or rather, not expressing its potential, in showing how this helps us understand climate change impacts on tropical species in fragmented landscapes. This can be improved by first giving more context (in the introduction) for how fragmentation can further disrupt the continuity of populations as they respond to shifting climates (possible predictions and well formulated questions). For example, will fragments experience local environmental changes the same way as large tracts of continuous forest? If not, how are they expected to differ? Rather, most of the introduction text focuses on stating that we don’t know much about these systems, which is valid, but there is little development beyond that. Adding more meat here to explore what we do know, about responses to climate change along other montane gradients, and/or responses to fragmentation in other systems, could set up the study better.

Second, I think the authors can do more to dig into the results, e.g., to ask if there are particular characteristics of species that are associated with movements (upslope or downslope, or the extent of shifts). Can we draw generalities on how tropical communities/groups of species/guilds will be impacted? Are there any comparisons that can be made across studies to draw additional insights here? I appreciate that brevity is emphasized, but it seems there is room to address this in more depth in the introduction and discussion, and doing so could make this a stronger contribution.

Third, I am left unconvinced regarding the fragmentation component of the study and the role it plays either formally or conceptually in the manuscript. There is not much depth in assessing how fragmentation affects how species shift in their ranges, except to say that the study was performed in a fragmented landscape. The results are considered in the extent of species range shifts compared to those found in other studies, but this is all done without thinking about any specific or general impacts of fragmentation, qualitatively or quantitatively. Would there be a way, for example, to test whether species track expected climate shifts more closely if they are found within the same forest blocks (or conversely, more likely to lag behind expected shifts if they are in different forest blocks)? It seems there could be a way to assess whether species that additionally have to contend with jumping between fragments in order to shift ranges are lagging further behind those that do not (i.e., the species that can move upslope within the same forest block). (It’s also possible that I don’t fully understand the layout and connectedness of the forest patches in the landscape – I’m just making inferences here based on Table 1).

As an aside, I found the patterns of slower colonization at the upper range limits really interesting. I wonder if there are other behavioral drivers that mediate these movements? Consider conspecific attraction – where individuals are more likely to settle in an area if conspecifics are already present (particularly if the presence of individuals serves as a measure of habitat quality). This could apply to your system, since individuals moving upslope from the lower limit are moving into areas where conspecifics are present, whereas those at the upper range limit are effectively colonizing areas where the species is all together absent or at most at very low densities. If conspecific attraction is important for colonization, this could create a lag in movements at upper limits versus lower ones. Is there any documentation of this elsewhere? It would be an interesting point to entertain briefly in the discussion, space permitting.

Line specific comments and recommendations are below:

Introduction

Lines 51: desvasting -> devastating

Lines 55-56: The authors point out here that we know little about how elevational range shifts may occur in the tropics when species are simultaneously responding to habitat discontinuities and fragmentation. This is an important theme that makes this study unique from others, but there is little other context for how this could compound responses to climate change. What potential impacts have been proposed? What will we learn by doing such studies in fragmented landscapes? I think the overall impact of the study will be higher if this is articulated more, here in the introduction, and connected with some of the ideas put forward in the discussion (or even better if fragmentation can be analyzed formally).

Lines 68-74: I encourage the authors to state some hypotheses clearly at the end of the introduction. What are you expecting to see with these surveys? Could you set up the comparison to expected changes due to warming, based on other studies? Having more context ahead of this, as recommended above, would better lead into the questions/aims of the study.

Line 59-60: Should these references also be numbered?

Methods:

Line 88: What is the “cold season” – is this characterized by less rain? Is this associated with the breeding season for the bird community? Notably, the resurvey is sampling a narrower duration of months compared to the original survey -- are the authors confident that these sampling periods are nevertheless capturing the same season? (In other words, could slight differences in seasonality be an issue here?)

Lines 111-161: I think the analytical methods are well executed here. I like the approach of using Monte Carlo simulations to account for differences in capture numbers between the time periods (though I had to read through this several times to get it), as well as the PCoA on the Bray-Curtis matrix.

Results:

Is there a way of explicitly considering the compositional change or range shifts within and between forest blocks? This seems important to fully base the study on the effects of fragmented forest on movement rates (otherwise the incorporation of fragmentation is a bit superficial, if there is no formal characterization in the study design or data analysis).

Lines 166-167: Which species were not represented in the resurvey? >13% decline in species richness is substantial. Could additional summaries be given for those species missing from the current survey?

Discussion:

In the discussion, I would like to see more in-depth evaluation of the kinds of species that move downslope, or those that are more likely to move upslope. Given the possibility for comparisons across three studies (Forero-Medina et al., Freeman & Class, and your study) it seems there is ample opportunity to dig into these patterns a bit more (even while maintaining brevity in the manuscript). I’ll reiterate here that this study makes an important contribution to our understanding of upper and lower range limit shifts in tropical systems, particularly by adding a new dataset from the Afrotropics. But the manuscript now stands more as a report on these patterns, yet it could do more to explore the context of patterns, first by setting up clearer questions and possible predictions at the beginning, and then layering some comparisons across studies that considers which species/groups have predictable responses (or not) in your study and across others like this one.

Lines 260-270: Here you report the greater frequency of movements within forest blocks than between forest blocks – so this points to dispersal limitation of species and/or sensitivity to cross gaps. Are these formally reported in the results section? (These kinds of results can put more emphasis on the interactions between fragmentation and climate change in affecting range shifts).

Lines 295-298: “Our results suggest that climate warming in combination with average distance among fragments and the limited dispersal ability of many species are very likely important contributory factors to the observed range contractions in bird species in the Usambara Mountains.”

Here I don’t really think the authors have completely made this case, especially as the results relate to the average distance among fragments. Distance among fragments was not considered explicitly in the analysis itself (I think it should be).

Lines 320 – 332: I like the emphasis on the high elevation endemics in drawing attention to the conservation priority for this region, though the discussion is necessarily limited since for most of the endemics there are few data available to do a robust analysis. Can the authors address (through personal observation) whether these endemic species are rare numerically, or just not easily detected using the methods employed in this study?

6. PLOS authors have the option to publish the peer review history of their article (what does this mean?). If published, this will include your full peer review and any attached files.

Reviewer #1: No

Reviewer #2: No

Reviewer #3: No

---

## [Author Response · Author response to Decision Letter 0]

16 Dec 2020

Please see the attached Response to Reviewers

---

## [Decision Letter · Decision Letter 1]

17 Feb 2021

PONE-D-20-31582R1

Afrotropical montane birds experience upslope shifts and range contractions along a fragmented elevational gradient in response to global warming

PLOS ONE

Dear Dr. Neate-Clegg,

Thank you for submitting your manuscript to PLOS ONE. After careful consideration, we feel that it has merit but does not fully meet PLOS ONE’s publication criteria as it currently stands. Therefore, we invite you to submit a revised version of the manuscript that addresses the points raised during the review process.

We look forward to receiving your revised manuscript.

Kind regards,

Bhoj Kumar Acharya, PhD

Academic Editor

PLOS ONE

Additional Editor Comments (if provided):

The authors have satisfactorily addressed all the concerns of the editor and three reviewers. The MS was sent to all the original reviewers and all of them are mostly satisfied with the revision. Reviewer 1 has provided some minor comments and I suggest the authors to address them carefully.

Reviewers' comments:

Reviewer's Responses to Questions

**Comments to the Author**

1. If the authors have adequately addressed your comments raised in a previous round of review and you feel that this manuscript is now acceptable for publication, you may indicate that here to bypass the “Comments to the Author” section, enter your conflict of interest statement in the “Confidential to Editor” section, and submit your "Accept" recommendation.

Reviewer #1: (No Response)

Reviewer #2: All comments have been addressed

Reviewer #3: All comments have been addressed

2. Is the manuscript technically sound, and do the data support the conclusions?

Reviewer #1: Yes

Reviewer #2: Yes

Reviewer #3: Yes

3. Has the statistical analysis been performed appropriately and rigorously? 

Reviewer #1: Yes

Reviewer #2: Yes

Reviewer #3: Yes

4. Have the authors made all data underlying the findings in their manuscript fully available?

Reviewer #1: Yes

Reviewer #2: (No Response)

Reviewer #3: Yes

5. Is the manuscript presented in an intelligible fashion and written in standard English?

Reviewer #1: Yes

Reviewer #2: (No Response)

Reviewer #3: Yes

6. Review Comments to the Author

Reviewer #1: Thanks to the authors for doing a great job with revisions. One very important point:

I like the addition of explicit consideration of elevational range changes in fragmented landscapes. However, and this is a big however, it does mean that the whole pitch of the paper has changed based on the results.

This is an example of hypothesizing after the results are known. The hypothesis here is that fragmentation differentially affects range shifts at lower vs. upper elevation limits via reduced rates of colonization. As is, a reader assumes that the study was designed to test this hypothesis. This is not the case.

To fix this, the authors should be open about the relationship between their data and hypotheses about fragmentation. The authors should clearly state (1) what their results were and (2) introduce the hypothesis as a possible explanation of their results [Rather than (1) introduce the hypothesis and (2) describe a test of these predictions]. For example, the authors can say they observed a difference in changes at low vs. high elevation limits towards the end of the Introduction, then subsequently introduce the idea that difficulty of upslope colonizations in fragmented landscapes is a possible explanation for this interesting observation. All text, and the conceptual figure, should stay. Just be clear that these are post-hoc hypotheses. This is perfectly valid, does nothing to detract from the paper, and correctly represents this project [as far as I can judge, based on the original manuscript].

A couple important points:

clarify in Abstract that landscape was already fragmented in the 1970s.

Suggestion: Add a figure showing changes in elevations for each species (e.g. each species a column, elevation on the y axis). And/or a strip chart / box plot showing change in total elevational size to illustrate that most species are contracting in range.

And the following thoughts:

text in Figure 1 low-resolution / hard to read. True for other figures too - make sure final version are high-res.

beyond the scope of this paper, but interesting to think about specific predictions for range shifts at species’ lower limits in fragmented landscapes. Figure 1 illustrates the situation if temperature is the driver (species disappears from lowest fragment and is not present at lowest elevation of second-lowest fragment). One possibility is that upslope shifts at species’ lower limits result from being “pushed” up by biotic interactions linked to temperature. In this case, and if dispersal is generally very low (not just for the birds, but also for species whom they interact with in ways important for range limits), then you might expect the species to disappear from the lowest fragment (where it is pushed up) but still live at the lowest elevation of the second-lowest fragment (where there is nothing arriving to “push” it upslope).

Reviewer #2: The authors have done what they can to address the problem of the separation of the East and West Usambaras by lower-elevation savannah. I know the distances among fragments are the same, but I still worry that few, if any of these species would cross lowland savannah between the two ranges. Nevertheless, the new analyses address this issue to some extent and the conclusions recognize how extremely limited these species are in their ability to move upslope, a problem exacerbated by the separation of the two mountain ranges. Therefore, I am now confident that the conclusions drawn from the data are fully supported.

Reviewer #3: I commend the authors in putting together such an interesting and well executed study. The authors were attentive to my suggestions and addressed the requested edits thoroughly. I like the addition of the conceptual model and predictions, the added consideration of effects across species with different natural history characteristics, and the broader comparisons to other tropical studies globally.

7. PLOS authors have the option to publish the peer review history of their article (what does this mean?). If published, this will include your full peer review and any attached files.

Reviewer #1: No

Reviewer #2: No

Reviewer #3: No

---

## [Author Response · Author response to Decision Letter 1]

22 Feb 2021

Please see attached response to reviewers

---

## [Editor Report · Decision Letter 2]

4 Mar 2021

Afrotropical montane birds experience upslope shifts and range contractions along a fragmented elevational gradient in response to global warming

PONE-D-20-31582R2

Dear Dr. Neate-Clegg,

We’re pleased to inform you that your manuscript has been judged scientifically suitable for publication and will be formally accepted for publication once it meets all outstanding technical requirements.

Kind regards,

Bhoj Kumar Acharya, PhD

Academic Editor

PLOS ONE

---

## [Editor Report · Acceptance letter]

19 Mar 2021

PONE-D-20-31582R2 

Afrotropical montane birds experience upslope shifts and range contractions along a fragmented elevational gradient in response to global warming 

Dear Dr. Neate-Clegg:

I'm pleased to inform you that your manuscript has been deemed suitable for publication in PLOS ONE. Congratulations! Your manuscript is now with our production department. 

Kind regards, 

on behalf of

Dr. Bhoj Kumar Acharya 

Academic Editor

PLOS ONE